# Synergistic Antitumor Effect of Combined Radiotherapy and Engineered *Salmonella typhimurium* in an Intracranial Sarcoma Mouse Model

**DOI:** 10.3390/vaccines11071275

**Published:** 2023-07-23

**Authors:** Zhipeng Liu, Sa-Hoe Lim, Jung-Joon Min, Shin Jung

**Affiliations:** 1Brain Tumor Research Laboratory, Biomedical Research Institute, Chonnam National University Hwasun Hospital, Gwangju 58128, Republic of Korea; jivincjl@gmail.com (Z.L.); sahoe@chonnam.ac.kr (S.-H.L.); 2Department of Neurosurgery, Chonnam National University Medical School, Hwasun Hospital, 322 Seoyang-ro, Gwangju 58128, Republic of Korea; 3Department of Nuclear Medicine, Institute for Molecular Imaging and Theranostics, Chonnam National University Medical School, Hwasun Hospital, 322 Seoyang-ro, Gwangju 58128, Republic of Korea; jjmin@jnu.ac.kr

**Keywords:** radiotherapy, immunotherapy, *Salmonella typhimurium*, intracranial sarcoma, adjuvant, neoadjuvant, mice

## Abstract

Intracranial sarcoma is an uncommon aggressive cancer with a poor prognosis and a high recurrence rate. Although postoperative adjuvant radiotherapy (RT) is the most recommended treatment strategy, it does not significantly improve survival rates. In this study, we used an attenuated *Salmonella typhimurium* strain engineered to secrete Vibrio vulnificus flagellin B (SLpFlaB) as an immunotherapy to assist with the antitumor effects of RT on intracranial sarcoma. In vitro, the expression of γH2AX and cleaved caspase-3 was analyzed by Western blot. In vivo detection of SLpFlaB colonization time in tumors was measured using an in vivo imaging system (IVIS). Tumor growth delay and elimination were demonstrated in an intracranial mouse model, and the distribution of macrophages, M1 macrophages, and CD8^+^ cells after treatment was measured using FACS analysis. Our findings in vitro suggest that combination therapy increases S-180 radiosensitivity, the expression of DNA double-strand breaks, and programmed cell death. In vivo, combination treatment causes intracranial sarcoma to be eliminated without tumor recurrence and redistribution of immune cells in the brain, with data showing the enhanced migration and infiltration of CD8^+^ T cells and macrophages, and an increased proportion of M1 macrophage polarization. Compared to RT alone, the combination therapy enhanced the radiosensitivity of S-180 cells, promoted the recruitment of immune cells at the tumor site, and prevented tumor recurrence. This combination therapy may provide a new strategy for treating intracranial sarcomas.

## 1. Introduction

Although radiotherapy (RT) as an adjuvant therapy to surgical resection in sarcomas shows local control, many patients develop distant brain metastases and have poor survival outcomes [1,2]. Tumors consist of a hypoxic core and a necrotic center, and the efficacy of RT mainly depends on the tissue’s oxygen content. The low oxygen concentration in the necrotic and hypoxic regions is a common cause of RT failure. Drug distribution is essential for the efficacy of chemotherapy. Poor tumor vasculature interferes with drug delivery, limiting chemotherapy efficacy, especially in necrotic and hypoxic regions [3]. The hypoxic areas and tumor vasculature are abnormal, rendering most cancer treatments ineffective. Due to the radioresistance of sarcomas, traditional treatment does not improve survival. Therefore, finding a new therapeutic strategy is essential to overcome CNS sarcoma.

RT has been used in 50% of cancer patients [4,5]. Historically, research on the mechanism of RT mainly focuses on the biological effects of delivering high physical energy of radiation to cancer cells, causing DNA damage in cancer cells and leading to cancer cell death [6]. Along with RT research, recent data indicate that radiation enhances antitumor immune responses by inducing tumor immunogenic cell death and the release of damage-associated molecular patterns (DAMPs), promoting antigen presentation, altering the tumor microenvironment, and transforming tumors from a “cold” to a “hot” tumor state [7,8,9,10]. In preclinical and clinical trials, RT was shown to activate vaccine responses in situ, aimed at transforming irradiated tumors into immunogenic centers, thus promoting innate and adaptive immunity [11,12,13,14]. However, some studies have shown that local recurrence often occurs after RT due to the local upregulation of the PD-L1/PD-1 axis after RT, which suppresses the radiation-induced immune response, thereby increasing the resistance of cells to DNA damage [15,16]. This recurrence suggests that the response induced by RT alone is not sufficient to maintain antitumor immunity. These studies provide the theoretical basis for combining RT and immunotherapy to enhance tumor-specific immune responses. Thus, the rational design of combination therapy with immunomodulators and RT is essential.

Immunotherapy for cancer treatment utilizes an entirely different approach to chemotherapy and RT by modulating the immune system to attack cancer cells rather than directly targeting the tumor [17]. Compared with other therapies, immunotherapy has two key properties: the immune system’s immune memory and the ability to detect and destroy tumor cell mutations as they arise [18]. In the current study, an engineered attenuated *Salmonella typhimurium* that initiates innate and adaptive immunity elicited interest. *Salmonella* attacks cancer in two ways. First, *Salmonella* targets the hypoxic and necrotic regions, accumulates around the tumor, infiltrates, and replicates within the tumor cells. Second, *Salmonella* can serve as a vehicle to deliver therapeutics to attack tumors [19]. *Salmonella* exhibits a remarkable ability to target the hypoxic and necrotic regions of tumors. Therefore, developing this cancer-specific immunotherapy might aid the shortcomings of conventional therapy.

Here, we used an engineered *Salmonella typhimurium* strain to secrete Vibrio vulnificus flagellin B (SLpFlaB) [20], which activates the immune response by stimulating toll-like receptor (TLR) 4 and 5 pathways. Combination therapy incorporating this approach as an adjuvant for RT saw tumor was inhibited; this combination therapy changed the distribution of immune cells in the brains of mice and promoted the apoptosis of sarcoma cells, confirming that combination therapy promotes immune responses and enhances synergistic antitumor effects.

## 2. Materials and Methods

### 2.1. Cell, Bacteria, and Animals

The mouse sarcoma S-180 cell line (KCLB, Seoul, Republic of Korea) was used for cell culture and maintained in Dulbecco’s Modified Eagle’s Medium (Gibco, New York, NY, USA) containing 10% fetal bovine serum (Gibco, New York, USA) and 1% penicillin–streptomycin (Gibco, New York, NY, USA) at 37 °C in a 5% CO_2_ environment. We obtained the engineered FlaB-expressing bacteria from Professor Min (Department of Nuclear Medicine, Chonnam National University, Hwasun Hospital). This SLpFlaB was maintained in a broth containing ampicillin (50 µL/mL) and kanamycin (50 μg/mL) and incubated at 37 °C at 200 rpm overnight. Female C57BL/6 mice aged 6–8 weeks old were purchased from (Orient Bio, Seongnam, Republic of Korea) and housed under specific pathogen-free conditions. All experimental procedures, including animal care, experimentation, and euthanasia, were carried out under an agreement approved by the Animal Research Committee of Chonnam National University.

### 2.2. Western Blotting

We used the Bicinchoninic Acid Protein Assay Kit (Thermo Scientific, Waltham, MA, USA) to determine the protein concentration of isolated protein samples. We separated the target proteins with SDS-PAGE and transferred them to a polyvinylidene fluoride membrane, which we immersed in a blocking buffer (5% skim milk in TBST) for two hours. The membranes were incubated overnight at 4 °C in primary antibodies. γH2AX (Cell Signaling, Massachusetts, USA), cleaved caspase-3 (Cell Signaling, Danvers, MA, USA), and β-actin (Santa Cruz Biotechnology, Heidelberg, Germany) antibodies were used to look at radiation-induced DNA damage, initiation and execution of apoptosis, and overall protein loading, respectively. The membrane was washed three times with TBST (five minutes each), then incubated with secondary anti-rabbit antibodies for one hour at room temperature. Chemiluminescence detection was conducted using the Immobilon Western chemiluminescent HRP Substrate (Merck KGaA, Darmstadt, Germany). We used an Amersham Imager 600 (GE Healthcare, Chicago, IL, USA) for figurative assays.

### 2.3. Intracranial Sarcoma Mouse Model

The scalps of the mice were shaved under anesthesia and wiped with iodine. Then, a 0.2 mm hole was drilled into the skull, creating an injection site 2 mm to the right of the bregma and 4 mm deep below the cortical surface. Next, 5 μL of DMEM containing 5 × 10^3^ S-180 cells at a rate of 1 μL/min was injected into the right striatum of the mice, using a stereotaxis (Gaonbio, Yongin, Republic of Korea), followed by cranial closure using bone wax, and then sutures [21].

### 2.4. Animal MRI and IVIS

To evaluate the status of SLpFlaB inoculated into mouse brain tumors, we anesthetized the mice with 2.7% isoflurane, 69% N_2_O, and 30% O_2_ and intraperitoneally injected 0.75 mg D-luciferin (Caliper Life Sciences, Waltham, MA, USA). Then, we placed the anesthetized mice in the light-tight chamber of the IVIS100 imaging system. Detection of firefly luciferase signals was performed using Living Image software v. 2.25 (Caliper Life Sciences, Waltham, MA, USA).

An animal magnetic resonance imaging (MRI) machine (M7TM compact MRI scanner, Shoham, Israel) was used to obtain mouse tumor images. Mice were anesthetized in 2.7% isoflurane, 69% N_2_O, and 30% O_2_, and we used a syringe with a 27-gauge fixed needle to inject 100 μL of enhancer (Magnevist, BAYER, Leverkusen, Germany). Five minutes later, the mice were placed in the MRI machine, and we performed a prescan to determine the location of the brain. We adjusted the parameters and positions of the sagittal, coronal, and horizontal planes to make the tumor site the target of the MRI imaging.

### 2.5. Radiation Treatment Schedule

In the in vitro study, S-180 cells were seeded in six-well plates at a density of 1 × 10^5^ cells and co-cultured with SLpFlaB (1:10) for three hours. After co-culturing, the cells were exposed to 8 Gy, and the medium was changed. The cells were harvested after being cultured for either 3 or 24 h.

In the in vivo study, MRI was performed on Day 7 after the intracranial injection of S-180 cells. Mice were divided into five groups: (1) control; (2) RT; (3) SLpFlaB; (4) adjuvant (SLpFlaB was administered after RT); and (5) neoadjuvant (SLpFlaB was administered before RT). All animals were anesthetized with 1.5% pentobarbital by intraperitoneal injection, and complete loss of consciousness was determined by toe pinch before RT. The SLpFlaB group received SLpFlaB 1 × 10^6^/5 μL intratumoral injection on Day 7. The RT group received 14 Gy irradiation on Day 7. The neoadjuvant group received SLpFlaB 1 × 10^6^/5 μL intratumoral injection on Day 4 and 14 Gy irradiation on Day 7. The adjuvant group received 14 Gy irradiation on Day 7 and SLpFlaB 1 × 10^6^/5 μL intratumoral injection on Day 10. After SLpFlaB treatment, L-arabinose was injected intraperitoneally once a day for a total of ten days to induce the expression of the therapeutic genes.

### 2.6. Isolation of Immune Cells from Mouse Brain

We anesthetized the mice with 1.5% pentobarbital sodium injected intraperitoneally, and we confirmed that they had reached the surgical plane by pinching the toes of the mice without any response. We next fixed the mouse on the surgical station and used tissue scissors to cut open the mouse’s chest skin and sternum in sequence to fully expose the heart. A small incision was made in the right atrium, and a 20 mL syringe was used to perfuse saline into the left ventricle slowly until the liver turned pale and the outflow from the right atrium was blood-free. The skull was carefully cut open using fine scissors, and the skull and brain were removed [22]. The brains were stored in Roswell Park Memorial Institute (RPMI) 1640 medium (Gibco, New York, NY, USA). To isolate a single brain cell, we used a sterile scalpel to cut the brain into 1–3 mm pieces and combined the brain fragments with type IV collagenase (100 U/mL; Gibco, New York, NY, USA) and DNase (100 μg/mL; Thermo Scientific, Waltham, MA, USA) at 37 °C in 5% CO_2_ for 40 min while shaking. We used 100 and 40 μm cell strainers (SPL, Pocheon, Republic of Korea) to filter single cells and remove the red blood cells, mixing them with 5 mL of 30% Percoll buffer (Sigma, St. Louis, MO, USA). We slowly infused the cells with 5 mL of 70% Percoll buffer, centrifuging them to harvest brain immune cells. These cells were collected in RPMI 1640 containing 10% fetal bovine serum and 1% penicillin–streptomycin at 4 °C [23].

### 2.7. Flow Cytometry

To detect the distribution of immune cells, we stained 1 × 10^6^ single cells with a Live/Dead Stain (1:1000; BD Biosciences, San Jose, CA, USA), Pacific blue-conjugated CD3 (1:200; BD Biosciences, San Jose, CA, USA), FITC-conjugated CD4 (1:200; BD Biosciences, San Jose, CA, USA), APC-conjugated CD8 (1:200; BD Biosciences, San Jose, CA, USA), APC-Cy™7-conjugated CD45 (1:200; Thermo Scientific, Waltham, MA, USA), FITC-conjugated CD11b (1:200; Thermo Scientific, Waltham, MA, USA), Pacific blue-conjugated MHC II (1:200; BioLegend, San Diego, CA, USA), and PE-conjugated CD86 (1:200; BD Biosciences, San Jose, CA, USA) antibodies for 30 min at 4 °C without light. The samples were processed on a BD FACSCanto II (BD Biosciences, San Jose, CA, USA), and all data were analyzed with FlowJo v10 software (Tree Star).

### 2.8. Statistical Analysis

All statistical analyses were performed with GraphPad Prism (GraphPad Software 9.0). One-way ANOVAs were performed for analyses across multiple groups. Unpaired *t*-tests were performed for tumor volume. The log-rank test was performed for survival data. A *p*-value of ≤0.05 was considered to be statistically significant (* *p* ≤ 0.05, ** *p* ≤ 0.01, *** *p* ≤ 0.001, **** *p* ≤ 0.0001). The results were expressed as the mean ± standard deviation (SD).

## 3. Results

### 3.1. Radiosensitization by Combination Therapy of S-180 Cell Line

In the Western blotting analysis, the effect of SLpFlaB combined with RT was confirmed by detecting levels of DNA double-strand breaks and apoptosis in tumor cells after treatment. There was an increase in the markers representing DNA double-strand breaks (γH2AX) and programmed cell death (cleaved caspase-3) after tumor cells were treated with RT alone, and this increase was more pronounced after combined treatment (Figure 1).

### 3.2. Safety and Tumor-Suppressive Effects of SLpFlaB

The intracranial tumor model is different from a subcutaneous model. In the intracranial mouse model, as tumor volume increases, body weight will continue to decrease. We found no significant difference in body weight between the four groups within 12 days after SLpFlaB infection (Figure 2A). However, there was a trend in which the body weight of the mice changed with the amount of SLpFlaB treatment. The weight loss of the mice was most evident in the SLpFlaB 1 × 10^7^ group, and the average survival time was significantly shortened. In the SLpFlaB 1 × 10^5^ and SLpFlaB 1 × 10^6^ groups, the weight loss of the mice was similar. However, the survival period of the SLpFlaB 1 × 10^6^ group was significantly prolonged (Figure 2B). The data show that SLpFlaB 1 × 10^6^ had no toxic effect on mice and had an apparent antitumor effect; therefore, in subsequent experiments, we used 1 × 10^6^ SLpFlaB as the primary therapeutic dose. Additionally, we found that SLpFlaB did not infect other tissues after inoculation and was continuously expressed for nine days (Figure 2C), indicating its ability to proliferate within the tumor, during which it can promote its antitumor effects.

### 3.3. The Radiosensitizing Effect of SLpFlaB in an Intracranial Sarcoma Mouse Model

Four days after tumor implantation, MRI detected the formation of tiny intracranial tumors in the brains of mice (Appendix A). We then performed neuroimaging once a week to detect tumor morphology according to the described treatment plan (Figure 3A). After Day 14 of tumor transplantation, the mice received animal MRI once a week to investigate the tumor volume under different treatment methods (Figure 3B). The tumor volumes were obtained by three-dimensional (3D) quantitation [24], and the two-month survival rates of the mice were analyzed (Figure 3C,D). The tumor size of the control group was significantly different compared with the SLpFlaB group (*p* = 0.0003), RT group (*p* < 0.0001), neoadjuvant group (*p* < 0.0001), and adjuvant group (*p* < 0.0001). On Day 30, the tumor size of the adjuvant group was significantly different compared with the RT group (*p* = 0.0420). These data correspond to the results for mouse survival. Mice exhibited prolonged survival following SLpFlaB and RT. SLpFlaB enhanced the average survival from 13 days in the control to 15 days in the SLpFlaB group (*p* = 0.0146), 31 days in the RT group (*p* < 0.0001), 41 days in the neoadjuvant group (*p* < 0.0001), and 44 days in the adjuvant group (*p* < 0.0001). Although the two single treatments improved the mice’s survival, none survived for two months. In contrast, the two combined treatments showed sound therapeutic effects. The cure rates were 33% in the neoadjuvant group and 40% in the adjuvant group.

To observe the therapeutic effect of combined therapy on mouse intracranial sarcoma more directly, brains were extracted, and the tumor size in each group was visually observed (Figure 3E). Tumor volume appeared to decrease after SLpFlaB treatment. RT had a tumor-suppression effect, delaying tumor growth. In the combination group, there was no tumor in the cerebral cortex of the mice as seen in MRI, indicating that SLpFlaB combined with RT exhibits a synergistic effect on tumor suppression.

### 3.4. Analyses of Immunogenicity

We confirmed the effect of RT combined with SLpFlaB therapy on the redistribution of immune cells in mice (Figure 4A,B). To examine the T cells, the distribution of macrophages, and phenotype changes of macrophages in the tumor microenvironment, we dissected the brains of mice five days after treatment, isolated immune cells, and stained them with antibodies. CD45^high^ and CD11b^+^ represented macrophages, CD45^+^, MHC II^+^, and CD86^+^ represented M1 macrophages, and CD3^+^, CD4^+^, and CD8^+^ represented T helper and cytotoxic T cells. FACS analysis showed that RT alone increased the number of macrophages (*p* = 0.0146) in the mouse brain but had no significant effect on M1 phenotype macrophages and T cells. Under the impact of SLpFlaB that had infiltrated tumors in the combined treatment group, the macrophage population (*p* = 0.0037), M1 phenotype macrophages (*p* = 0.0009), and CD8^+^ T cells (*p* = 0.0109) were significantly increased. The results showed that SLpFlaB induced the infiltration of CD8^+^ T cells and macrophages into tumors and promoted the polarization of macrophages to the M1 phenotype. The combination therapy altered immune cell distribution in intracranial tumors of mice, inhibiting irradiated tumor growth.

### 3.5. Combination Therapy Prevents Tumor Recurrence

Considering the refractory and poor prognosis of the tumor, we continued to observe the cured mouse brain by MRI up to 150 days later (Figure 5). It is worth noting that there was no tumor recurrence in mice with cured tumors (regression) in the neoadjuvant and adjuvant groups, and there were no sequelae. These results indicate that RT combined with SLpFlaB treatment cured brain tumors in mice, prolonged their survival, improved their prognoses, and prevented the repopulation of tumor cells.

## 4. Discussion

Recently, a bacterial cancer treatment (BCT) strategy has become an emerging research direction. BCT activates the immune system and delivers therapeutic agents to the tumor site to kill tumor cells [25,26]. However, monotherapy is usually insufficient to cure cancer [27,28]. Therefore, we used SLpFlaB combined with RT as a treatment and demonstrated the importance of this strategy for cancer treatment.

In our study, we delivered SLpFlaB to the tumor site by intratumoral injection. The intratumoral injection overcomes the issue of blood–brain barrier permeability, and dosage is controllable while ensuring safety and low toxicity, maximizing the immune response of SLpFlaB, and avoiding the risk of systemic toxicity.

SLpFlaB alone showed the ability to prolong survival but not eliminate tumors, which is consistent with the results of other studies [28]. This effect occurs possibly because the immune environment of the brain is different from other sites; the brain has multiple restrictive immune responses and immune mechanisms that promote treatment resistance. Most peripheral immune cells are excluded from the CNS [29,30,31], which might allow intracranial tumors to be more aggressive and progress faster than other tumors.

Given the goal of improving post-treatment remission, we investigated the effects of SLpFlaB as adjuvant and neoadjuvant therapies for RT. The two treatment regimens showed unexpected tumor-suppressive results; the CNS sarcoma completely disappeared. More survival benefits were seen in the adjuvant group, perhaps because the tumor cells were more susceptible to SLpFlaB infection after RT [32,33]. In the neoadjuvant group, the lesser survival benefits observed might be due to impaired biological functions of SLpFlaB after RT, affecting any immune effect [34]. The exact mechanisms require further research.

To study the effect of SLpFlaB and RT on tumor cells, we detected the expression of γ-H2AX and cleaved caspase-3. The results showed that the combined treatment significantly increased DNA double-strand breaks and apoptosis in tumor cells, which suggests that combination therapy increases the radiosensitivity of tumor cells. Further observation of the expression of immune cells at the tumor site after treatment demonstrated that blood-derived macrophages were increased in the tumor tissue. The anti-inflammatory M2 phenotype macrophages shifted to the pro-inflammatory M1 phenotype. CD8^+^ T cell numbers were also increased. These results demonstrate that this treatment modifies the tumor microenvironment in the brain and activates an antitumor immune response. Furthermore, there was no recurrence after tumor elimination, indicating that this combination therapy triggered immune memory, resulting in long-term protection against metastasis and recurrence.

In our pre-experiments, treating 1 × 10^6^ of SLpFlaB alone to mice did not effectively stimulate the expansion of CD4 T cells and CD8 T cells in the tumor, which is consistent with the other research results; for example, in mice treated with 1 × 10^7^ of SLpFlaB, no expansion of CD4 T cells and CD8 T cells was detected [20]. But in another study, the authors treated 1 × 10^8^ of SL3261AT to mice, which effectively stimulated cytotoxic CD8 T cells, which could directly recognize and kill tumor cells [35]. The different dosages of *Salmonella* may cause a difference in the results. However, we cannot increase the dosage of SLpFlaB because of severe toxic effects in mice, especially the sensitive tissue of the brain. This further evidence that SLpFlaB is an effective RT adjuvant, and this combination therapy strategy increases the radiosensitivity of tumor cells, enhances the therapeutic effect of RT, effectively activates the immune response, and breaks the environment of immune tolerance. Our study does not fully explain how this combination therapy eliminates intracranial sarcomas because the synergistic antitumor effect of combined RT and SLpFlaB is multifaceted; this synergistic effect may be induced via a variety of pathways. For instance, RT was shown to cause tumor cells to embrittle and aggravate hypoxia in the tumor area, prolong the colonization time of *Salmonella typhimurium* in the tumor, induce changes in capillary permeability temporarily disrupting the blood–brain barrier, and facilitate the delivery of blood-derived immune cells to the CNS tumor site [36,37,38]. RT was also shown to induce tumor immunogenic cell death and enhance SLpFlaB to exert an immune response [39]. LPS can induce mtDNA synthesis and release, which is recognized by RAD50 in macrophages, and through the activation of STING and NF-κB signaling pathways, promotes the release of pro-inflammatory cytokines and the production of ROS, thereby enhancing radiosensitivity [40,41]. Also, LPS can promote mitochondrial dysfunction, selectively downregulating PD-L1 expression and increasing immune cell infiltration [42,43,44]. Overall, our study confirmed the effectiveness of this combination therapy. Future studies should verify why the combination of RT and SLpFlaB has unexpected antitumor effects.

## 5. Conclusions

More and more evidence suggests that RT can stimulate local and systemic immune responses through multiple mechanisms; this response may support tumor cell repair or promote tumor cell death. For example, the radiation dose and the number of fractions can have diametrically opposite results [45]. With the progress of research on the biological effects of RT, the combination of RT with immunotherapy will become increasingly of interest, and finding a more reasonable and effective combination therapy is an inevitable strategy to improve the outcome of cancer treatment [46,47]. This study provides an effective combination therapy strategy to demonstrate that the combination of RT and SLpFlaB has a synergistic antitumor effect in an intracranial sarcoma mouse model. The combination therapy enhanced the radiosensitivity of S-180 cells, promoted the recruitment of immune cells at the tumor site, and prevented tumor recurrence. In addition, we found that the adjuvant group showed a better antitumor effect than the neoadjuvant group, thus demonstrating the importance of treatment timing on antitumor responses and providing evidence for the design of rational therapeutic combinations to improve the antitumor effects.

## Figures and Tables

**Figure 1 vaccines-11-01275-f001:**
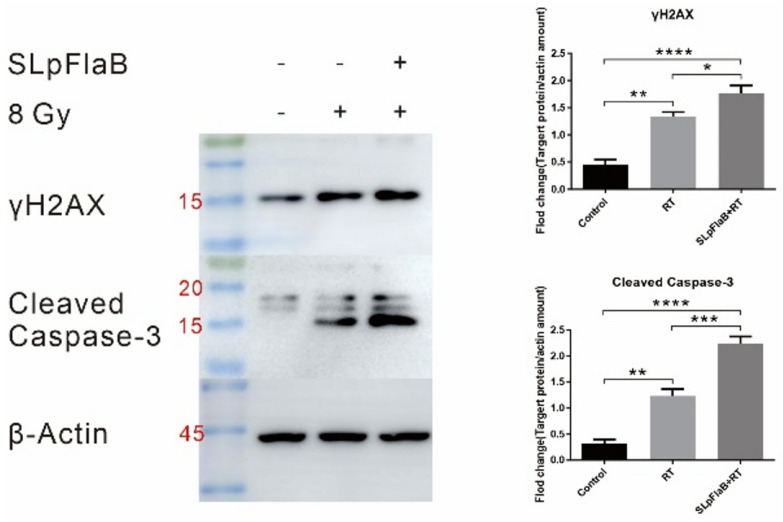
Effect on radiosensitivity of S-180 cell line after treatment with or without RT and SLpFlaB. Representative Western blot images showing the expression of γH2AX (DNA double-strand break), cleaved caspase-3 (programmed cell death), and ß-actin in the S-180 cell line. Quantification of Western blot analysis summarized by bar charts as the mean ± standard deviation (SD). * *p* ≤ 0.05, ** *p* ≤ 0.01, *** *p* ≤ 0.001, **** *p* ≤ 0.0001.

**Figure 2 vaccines-11-01275-f002:**
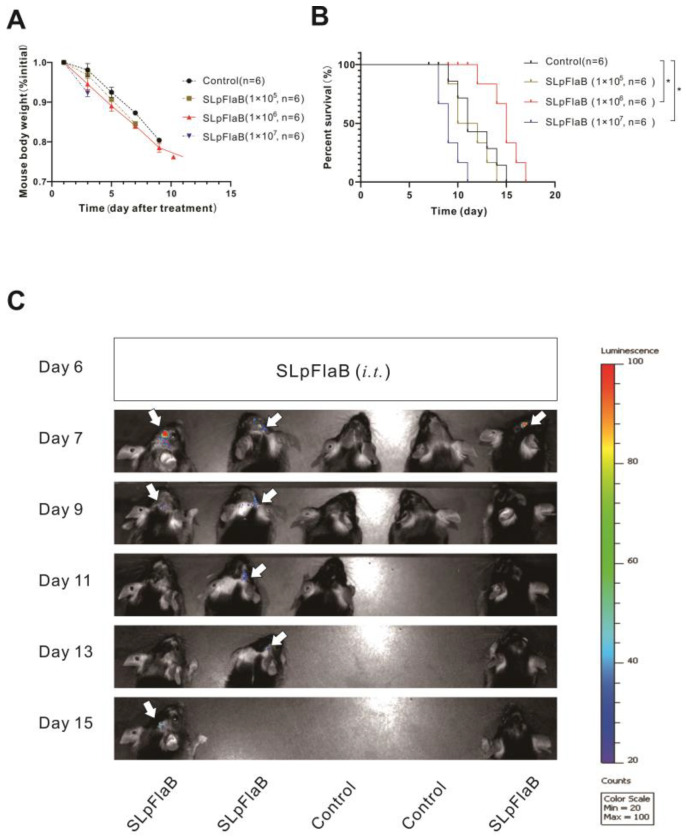
Examining the toxic effects, survival benefits, and colonization duration of an intratumoral injection of SLpFlaB. (**A**) Observed changes in C57BL/6 mice (*n* = 6) body weight after inoculation with different SLpFlaB numbers (1 × 10^5^/5 μL, 1 × 10^6^/5 μL, and 1 × 10^7^/5 μL) on Day 6. (**B**) Kaplan–Meier survival curve plots of intracranial tumor mice after inoculation with different SLpFlaB numbers (1 × 10^5^/5 μL, 1 × 10^6^/5 μL, and 1 × 10^7^/5 μL). (**C**) The bioluminescence expression was determined after inoculation with SLpFlaB (1 × 10^6^/5 μL) by optical imaging (IVIS) (*n* = 3). White arrows indicate SLpFlaB. All data are shown as the mean ± standard deviation (SD). * *p* < 0.05.

**Figure 3 vaccines-11-01275-f003:**
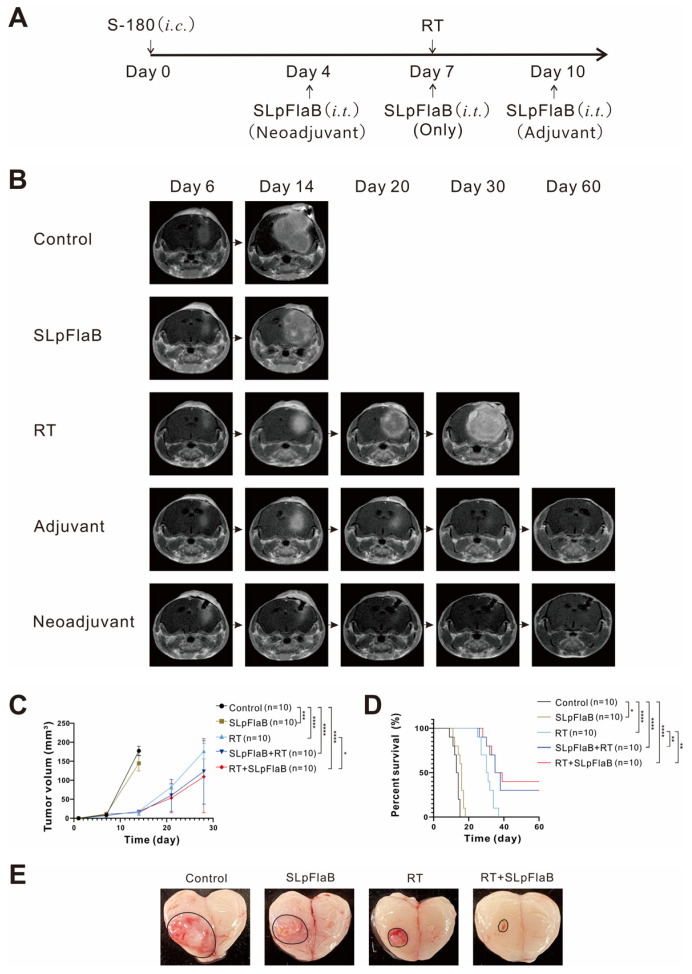
Experimental treatment regimens and efficacy of combination treatments. (**A**) Overview of the experimental timeline (*n* = 10). Adjuvant (SLpFlaB was administered after RT): neoadjuvant (SLpFlaB was administered before RT). (**B**) In vivo contrast-enhanced MRI T1 mapping of intracranial sarcoma mouse models. When mice in the same group were dead, the following MRI imaging was stopped. (**C**) The three-dimensional (3D) ROI-based quantitative measurement obtained brain tumor volume from each MRI slice. (**D**) Kaplan–Meier survival curves were used to estimate the survival of the S-180 intracranial mouse tumor model following the different treatments. (**E**) Tumor ex vivo photos on Day 14. Data are summarized by bar charts as the mean ± standard deviation (SD). Control, no treatment group; RT, radiation therapy (14 Gy); SLpFlaB, 1 × 10^6^/5 μL; adjuvant, SLpFlaB was administered after RT; and neoadjuvant, SLpFlaB was administered before RT. * *p* ≤ 0.05, ** *p* ≤ 0.01, *** *p* ≤ 0.001, **** *p* ≤ 0.0001.

**Figure 4 vaccines-11-01275-f004:**
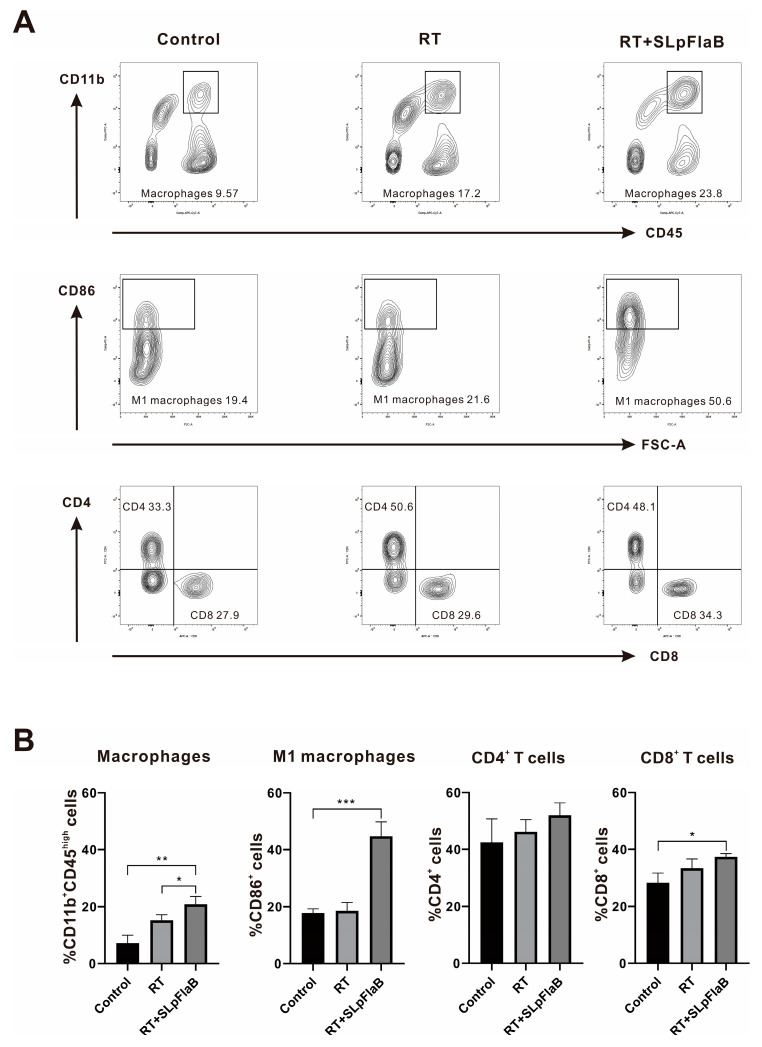
Flow cytometry confirmed the distribution of immune cells in the brain of the intracranial tumor model (*n* = 3). (**A**) Absolute numbers of each immune cell; macrophages (CD45^high^ and CD11b), M1 macrophages (CD45^+^, MHC II^+^, and CD86^+^), T helper cells (CD3^+^ and CD4^+^), and cytotoxic T cells (CD3^+^ and CD8^+^). (**B**) Quantification of each type of immune cell. Data are summarized by bar charts as the mean ± standard deviation (SD). Control, no treatment group; RT, radiation therapy (14 Gy); RT+ SLpFlaB, adjuvant: SLpFlaB was administered after RT; * *p* ≤ 0.05, ** *p* ≤ 0.01, *** *p* ≤ 0.001.

**Figure 5 vaccines-11-01275-f005:**
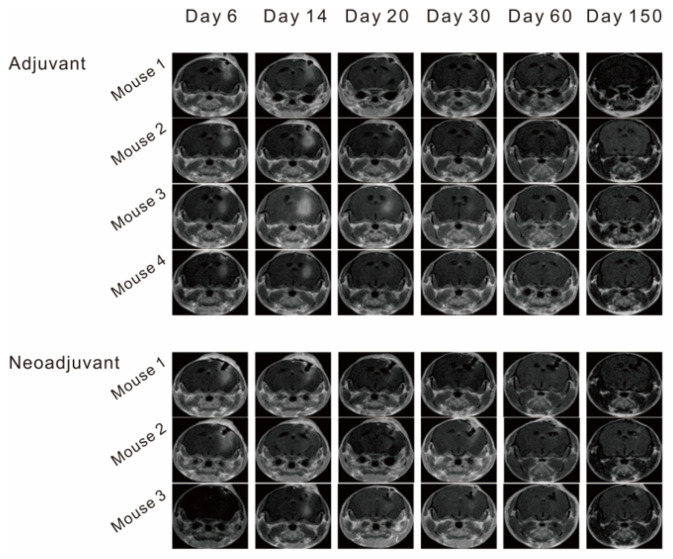
Assessment of cured mice using contrast-enhanced MRI T1 mapping on Day 150. Neoadjuvant (*n* = 3), adjuvant (*n* = 4).

## Data Availability

Not applicable.

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
