# Peer review of "Synergistic Antitumor Effect of Combined Radiotherapy and Engineered Salmonella typhimurium in an Intracranial Sarcoma Mouse Model"

_vaccines, 2023, doi:10.3390/vaccines11071275_

Round 1
Reviewer 1 Report
The manuscript “Synergistic antitumor effect of combined radiotherapy and engineered Salmonella typhimurium in an intracranial sarcoma mouse model “ by Liu et al have shown the antitumor effects of radiotherapy with attenuated salmonella. Several issues should be addressed.
1. It is difficult to maintain the culture of salmonella that could activate TLR4 and 5 in clinic. It would be valuable to evaluate whether salmonella is superior to TLR ligands such as LPS in this system. Otherwise, it is difficult to confirm the superiority of salmonella against more simple adjuvants. Please perform a small experiment just to compare salmonella and TLR ligands (only LPS would be fine).
2. I cannot obtain any information from Figure 2C. Please present comprehensible figure.
3. How is it possible to perform intratumoral injection on Day 4 with 5E3 tumor cells? Is it visible?
4. In figure 4, SLpFlaB alone group should be included. Also, no explanation whether this is adjuvant or neoadjuvant treatment.
5. Was the assessment of CD86 expression examined in CD11b population?
6. It is advised to deplete CD8 T cells to confirm that CD8 T cells play a role in this treatment.
N/A
Reviewer 2 Report
The manuscript entitled "Synergistic antitumor effect of combined radiotherapy and engineered Salmonella typhimurium in an intracranial sarcoma 3 mouse model" by Liu et al. synergism of radiotherapy and Salmonella typhimurium for treatment of sarcoma. The manuscript is well written and organized in scientific manner. The manuscript is suitable for publication in the journal following minor revision.
1. Introduction: First sentence is too long should be revised for simplification and better understanding.
2. The acronyms should be described in full when appearing first in the manuscript. Revise manuscript thoroughly for acronyms used.
3. References should be cited for procedures and protocols adopted until developed by self in the study.
4. The use of space between unit and value need to be uniform.
5. Some typographical mistakes like N2O, O2, etc.
6. Conclusion need to be modified and make more informative.
Reviewer 3 Report
In this research, the authors constructed synergistic antitumor effect of combined radiotherapy and engineered Salmonella typhimurium in an intracranial sarcoma mouse model. In my opinion, the current stage of this paper could meet the requirements of Vaccines after major revisions.
My comments are as details:
1. The quality of the figures is poor. The font size of the figures is too small. The authors should better improve it.
2. In Line 46-57, the authors should better point out the acquired resistance especially increased PD-L1 expression after RT, which may impair its efficacy. Some references could be added to this part, including 10.1002/advs.202207608.
3. Some minor mistakes exist in this paper, such as (1×105, 1×106, 1×107). The authors should better revise it.
4. In Fig. 2C, the signal of the bioluminescence expression was determined after inoculation with SLpFlaB by optical imaging (IVIS) was poor to see or read to get any information.
5. How was the immune status or how immune may affect the efficacy of combined radiotherapy and engineered Salmonella typhimurium should be discussed in the discussion. Some references could be added to this part, including 10.1016/j.ijbiomac.2022.10.167.
6. The figure legend was poorly written. The authors should better improve it.
Round 2
Reviewer 1 Report
The authors have answered to several of my concerns.
Minor comments are as follows;
1. Please show a photo of a small tumor after using 5×103 S-180 cells to plant intracranial tumors for four days as a supplementary data.
2. If the authors think CD8 T cells are not responsible for this treatment, please mention this issue in the discussion.
The quality of English in the answer letter is somewhat poor.
Reviewer 3 Report
The current version of this manuscript could be accepted.
Author Response
Thank you again for reviewing our manuscript and your comments, which have greatly helped us complete the manuscript.